# Business Model of Peer-to-Peer Energy Trading: A Review of Literature

**Hani Muhsen** [1,2,*] , **Adib Allahham** [3] , **Ala'aldeen Al-Halhouli** [1,2] , **Mohammed Al-Mahmodi** [2] , **Asma Alkhraibat** [2] and **Musab Hamdan** [2]

1   Department of Mechatronics Engineering, School of Applied Technical Sciences,
    German Jordanian University, Madaba 11180, Jordan; alaaldeen.alhalhouli@gju.edu.jo
2   Smart Grid Lab, German Jordanian University, Madaba 11180, Jordan;
    mohammed.almahmodi@gju.edu.jo (M.A.-M.); a.alkhraibat@gju.edu.jo (A.A.);
    Musab.hamdan@gju.edu.jo (M.H.)
3   School of Engineering, Newcastle University, Newcastle upon Tyne NE1 7RU, UK;
    Adib.Allahham@newcastle.ac.uk
*   Correspondence: hani.mohsen@gju.edu.jo

**Abstract:** Peer-to-peer (P2P) energy trading is a promising energy trading mechanism due to the deployment of distributed energy resources in recent years. Trading energy between prosumers and consumers in the local energy market is undergoing massive research and development, paying significant attention to the business model of the energy market. In this paper, an extensive review was conducted on the current research in P2P energy trading to understand the business layer of the energy market concerning business model dimensions: bidding strategies and the market-clearing approach. Different types of game theoretical-based and auction-based market-clearing mechanisms are investigated, including a detailed classification of auctions. This study considers the possibility of employing the P2P technique in developing countries and reviewing existing business models and trading policies. The business layer of the P2P structure plays a vital role in developing an effective trading mechanism based on interactive energy markets.

**Keywords:** peer-to-peer (P2P) energy trading; business model; bidding strategies; auctions; game theory; developing countries

## 1. Introduction

The energy market has been revolutionized in recent years due to various aspects. These aspects include the proliferation of distributed renewable energy resources and the increase in the number of prosumers in the local electricity network, as well as the development in information systems. The local energy market is undergoing several developments to accommodate energy participants through various trading mechanisms and balance the supply and demand within the microgrid. Prosumers who are the agents with renewable energy resources such as rooftop solar panels are integrated with the grid and trade their excess energy using certain policies that vary from one country to another, for example, net-metering and feed-in tariff [1]. An alternative policy is needed to adapt to the rapid development in the energy sector worldwide [2]. One of the promising trading strategies is P2P.

The term "peers" in the energy market indicates the grid-connected parties which could be two or more. These parties are available in the grid in two forms, ordinary consumers and prosumers (consumers who have their renewable energy resources such as rooftop solar systems). Peer-to-peer (P2P) energy trading enables prosumers to actively participate in the electricity market, and gain incentivized profits. The P2P market is constructed in the low voltage network within a microgrid or among neighbor microgrids. The article [3] presented the architecture of P2P market into four different layers: power

grid layer, information and communication technology (ICT) layer, control layer, and business layer. The business layer is an essential part of the P2P energy market that concerns maximizing participants' revenue. It includes bidding or pricing methodologies and market-clearing techniques. An online platform matches the asks and bids and clears the market according to the selected business model.

Several pilot P2P projects have been carried out across the globe, such as Piclo in the United Kingdom, Transactive Grid in the US, and PeerEnergyCloud in Germany [4,5]. The main developed P2P energy trading projects are compared in [6], and Zhou et al. [5] provided a detailed summary of these projects. All installed P2P projects are similar in platform design, but they have different business models. The business model refers to the process of matching and energy exchanging in the trading market, including bidding strategy, market clearing, and final settlement [7]. Three different types of energy markets models were traditionally used in energy trading, as stated in [8]: Pool market, Bilateral market, and Balancing market. According to [9], the business model structure in P2P can be found in three distinct kinds, according to the type of participants in the trading market: Business-to-Business (B2B), Customer-to-Customer (C2C), and Business-to-Customer (B2C).

Previous reviews of P2P technology focused on several aspects of the field. For instance, summarizing and comparing the existing projects in terms of similarities and differences as found in [5,6,10]. Further, the challenges and opportunities of this technique concerning the platform design and blockchain are reviewed in [1,4,11]. Other reviews take into consideration specific topics such as community-based market design [9], some market-clearing approaches [12], and the impact of applying the local energy market on the electricity network [13,14].

Business models are introduced in literature with countless articles highlighting diverse pricing and market design types. However, to the best of the authors' knowledge, there is no recent comprehensive review introducing the distinct types of business models in P2P energy trading, including bidding strategies and market clearing approaches. So, the main contributions of this paper can be summarized as follows:

- This paper reviews the innovative trading mechanisms used in the P2P energy market to investigate its efficiency and applicability.
- It intensively surveys the main dimensions of the business model: market-clearing approaches and bidding strategies that researchers used to build a business layer of local electricity exchange between peers.
- Reviewing the P2P trading policies of developing countries is considered a critical issue due to very constrained policies in the energy sector and insufficient articles and reviews of such innovative trading mechanisms.
- This review covers a comprehensive classification aspect of the auction methods applied in several sectors, especially in energy trading.
- It offers recommendations for future work directions for business model development and energy trading policies in developing countries.

The paper's structure is organized as the following: Section 2 presents the electricity market development; the local market structure is presented in Section 3; trading algorithms including bidding strategies, and market-clearing approaches are discussed in Section 4; game-theoretic approach and auction-based approach are discussed in Sections 5 and 6, respectively; business models in developing countries are discussed in Section 7 and the future work directions and recommendations for developing countries are presented in Section 8.

## 2. Electricity Market Development

Distributed Energy Resources (DER), which refer to different small-scale power generations such as rooftop solar systems and small-scale wind turbines, have rapidly increased in the electricity network. The increasing penetration of DERs leads to finding appropriate policies to incentivize residents and companies to take part and invest in renewable energy resources. The increasing penetration of DERs leads to finding appropriate policies to

incentivize residents and companies to take part and invest in renewable energy resources. Every nation develops distinct pricing methods and engages DERs with grid, and several trial platforms are designed and tested to accommodate P2P as the approach of the future energy exchange.

## 2.1. Existing Policies

Nowadays, different installed DERs are integrated with the grid, and their surplus energy can be exchanged with the network through different policies. The most commonly used policies are net metering (NEM) and feed-in Tariff (FiT) [15]. In FiT, the surplus energy of the prosumer can be sold to the retailer at an export price, which is mostly lower than the retail price (the price of buying electricity from the grid). While in NEM, the surplus energy at the prosumer's end is transferred to the grid so that it is reimbursed in other months of the year when the energy produced is insufficient to cover the prosumer's demand [16]. In addition, different policies are also used in different countries such as Feed-in-Premiums (FiPs), Generation Based Incentives (GBI), Supply Agreement with Renewable Energy (SARE), Large Solar Scale (LSS), Self-Consumption (SELCO), and tax credits/incentives [4,17].

## 2.2. Current Research Directions of P2P

Various research projects are directed towards P2P energy trading as a promising trading policy. The P2P energy market is introduced from several aspects to optimize the system performance by increasing scalability, minimizing power losses, evaluating the impact of transaction process on the network, and investigating distinct types of business models.

First, the scalability barrier of the P2P market is a critical issue to be handled while applying such a system in a large-scale market. This problem, as well as the possibility of integrating the current market and its regulations with the P2P market, are presented in [10]. The article [18] offers a method of clustering for optimizing the scalability of P2P marketplaces by the method of adaptive segmentation. Finally, in [19] a P2P scalable market design introduces a real-time and forward market, including bilateral contracts between conventional suppliers, intermediaries, and agents with renewable resources.

Second, electrical system losses' minimization is another direction of research interest in the P2P market. Guerrero et al. [20] designed a P2P energy system that prioritizes the closer customers with limited intervention of DSOs. This paper concerns evaluating the performance of the proposed mechanism that considers the distance to generate the list of preferences. The results illustrate that the driven mechanisms of electrical distance reduced the losses of the network compared to the forefront P2P mechanisms. A new market-clearing method that considers the power loss, actors' privacy, and utilization fees is proposed in [21]. When considering the network's fees, the user's decision depends on the costs that the supplier offers and the supplier's electrical distance. In [22], the article analyzed the impact of three phase unbalanced systems on P2P transactions and provided measures for the issue of loss allocation of the transactive energy TE by matching the physical parts of the radial distribution system with P2P energy trading.

Third, it is considered that P2P energy trading should benefit both the electricity network structure and ensure participants' profits. Guerrero and colleagues [23] proposed a method that considers the distribution-level network constraints of P2P trading, introducing a method to evaluate the P2P transaction impacts on the network and ensuring that it does not violate the restrictions of the network. The paper [24] aims to guarantee the benefits of both prosumer and consumer by figuring out the lowest and highest price of the traded electricity, by creating a policy of self-consumption and buying electricity for a beneficial P2P energy trade, which is based on the energy consumed each month, in which prosumers and consumers are matched. Another study [25] investigates the P2P transaction effects on the network and proposes a permission structure to make the settlements of several arrangements of P2P possible, without affecting the network operation.

Other studies focus on developing the market efficiency, for instance the economic impact of transforming customer participating tariffs from Energy-based tariff (EBT) to Power-based tariff (PBT) in P2P is proposed in [26], concluding that annual power tariff with trading registered the lowest costs in three categories when PBT is applied. Another study concerning cost allocation strategy, to study the impact of the decentralized market so-called exogenous approach, is proposed in [27]. It determines the method of estimating network charges. This approach can limit the stress that the market puts on the grid. The network charges allow the system operator to collect participants' charges, aiming to recover the cost. According to [28], market efficiency can be improved by using community energy storage (CES). The increase in the market's efficiency depends on how many trading partners are allowed.

## 3. Local Market Structure

The electricity network has witnessed continuous developments throughout the past few years in power flow and energy exchange between central utilities, consumers, and distributed energy resources (DERs). As a result, the conventional energy market is increasingly developing into more decentralized or distributed frameworks in order to maximize reliability and the level of security [18]. The article [12] presents current works on local energy market designs' prospects. Based on the level of decentralization and the method of engaging DERs in the network, the local electricity market can be classified into three different structures: community-based, full P2P, and hybrid market [9,12,29,30]. The key difference between the three types of market structure is that in full P2P market peers trade energy directly without a mediator. In contrast, mediators or aggregators are needed in the community-based markets to organize the trading process. In the hybrid market, the peer can choose whether to trade with other peers directly or through a mediator. Figure 1 represents three market structures: full P2P, hybrid market, and community-based in a, b, and c, respectively.

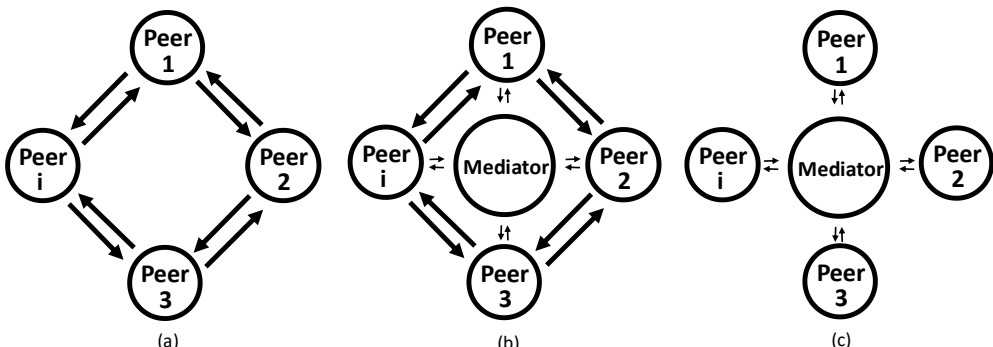

**Figure 1.** Market structures; full P2P (**a**); hybrid market (**b**); and community-based market (**c**).

A community-based market refers to energy trading among peers through a community manager or aggregator who receives data from participants, organizes selling-buying deals, and balances supply and demand in the network. This design is considered the most straightforward implementation in which trading is done through a coordinator [12]. Moreover, the community-based market seems to be a promising business model for the future P2P energy market implementation in the world, according to [31].

The second market design is complete P2P trading, the case of exchanging energy bilaterally between two peers, prosumers, or communities, directly without needing coordination from a third party [32]. Finally, the third category of market frameworks is the hybrid market, a combination of full P2P and community-based markets when users can decide whether to trade energy via a community coordinator or directly trade with neighbors [30].

Several aspects of P2P energy trading are discussed in [33], including a detailed classification of the market design concerning the advantages and disadvantages of each

distinct type. All centralized, decentralized, and distributed market designs are considered, concluding that the centralized has less uncertainty in the generation and demand of the peers due to the availability of a coordinator who controls the operational status of the peers. However, decentralized markets outweigh centralized in terms of privacy, scalability, and peers' freedom of contracting. The distributed structure combines the advantages and disadvantages of both previous designs.

Sousa et al. [9] introduced a comprehensive review of the community-based structure and compared the earlier three classifications of market paradigm, concluding that the hybrid design is the most appropriate one. In this infrastructure, agents can be grouped as separated communities and linked together, offering a lower communication cost and more scalable systems than the other two types. On the other hand, the convergence of the full P2P type has registered slow rates, and it demands costly communication networks because peers negotiate independently without a mediator [30].

Energy collective for the community-based market framework is presented in [34]. Energy collective is a number of prosumers who cooperate to perfect their energy utilization in a distributed market structure, and a supervisory node performs the collective agreements to set up a new local market for P2P trading. Morstyn et al. [19] introduce a real-time and forward framework with bilateral contract networks to directly involve prosumers in the market. Moreover, it presents this framework in energy contracts between conventional generators, prosumers with flexible and inflexible loads, and intermediaries.

A new local market is proposed in [35] for a distribution system in which P2P energy trading is integrated with margin and location pricing as an alternative to conventional pricing. It also proposed a novel pricing strategy for distribution system operator DSOs that depend on a day-ahead estimated cost, which is sent to the prosumer unidirectionally.

Electricity markets have been built with or without intermediaries. Mengelkamp et al. [36] present a market without needing central intermediaries. At the same time, in [26], it is assumed that a central governing authority that the participants could form takes the responsibility of conducting the bidding, trading, and clearing instead of the participants, and the benefits are distributed accordingly. They also derive seven optimal and efficient energy market components, proposing a microgrid named "Brooklyn" but excluding "Regulation" because it is still not allowed in most countries. Another trading platform is named "Elecbay" which is presented in [3], including a business model for the grid-connected microgrid.

## 4. Trading Algorithms

Several studies have been conducted to maximize participants' profits, incentivizing them to cooperate as prosumers in the local energy market. Further, achieving electricity demand-supply balance is another aim of setting up P2P frameworks. To this end, various policies and trading mechanisms have been presented in the literature concerning market structure, bidding strategies, and market clearing mechanisms. Figure 2 illustrates the three trading phases of the business layer: bidding, market clearing, and settlement.

### 4.1. Bidding Strategies

The P2P energy market is an interactive environment that enables participants: prosumers, consumers, and utilities to offer their asks and bids in order for them to maximize their profits. Several bidding strategies are developed in the literature to suit P2P energy trading in the local energy market. Many research articles studied distinct types of bidding strategies in the electricity market to investigate the opportunity of installing a P2P trading market. Article [37] introduces several bidding strategies and investigates their impacts on the conditions of the market. Table 1 illustrates several types of bidding strategies used in current studies of P2P energy trading.

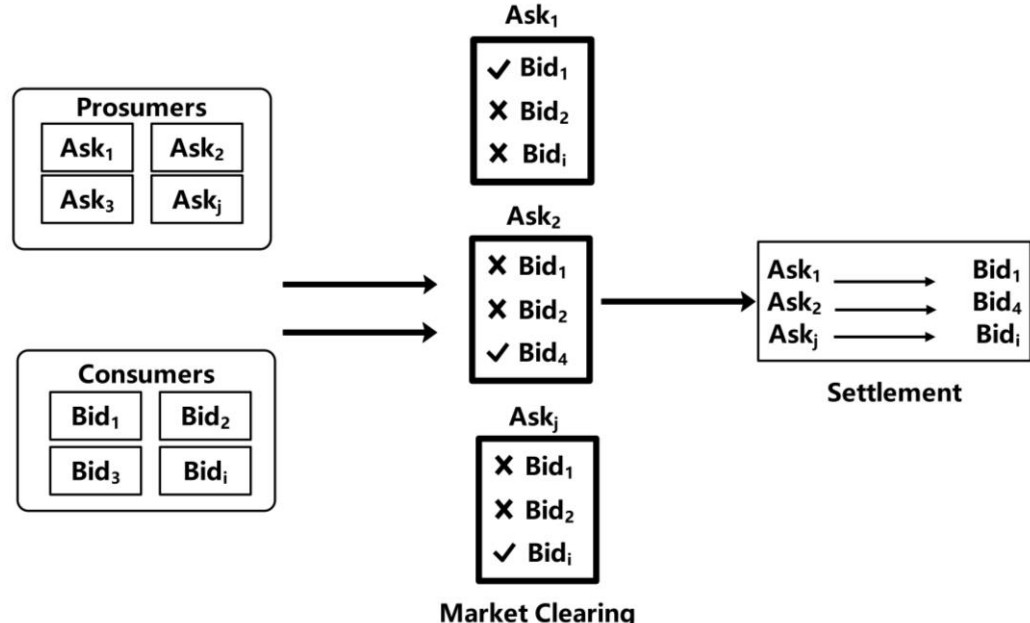

**Figure 2.** Trading phases of the business layer.

**Table 1.** Several types of bidding strategies that are used in current studies.

| Bidding Strategy | Definition | References |
|---|---|---|
| Zero intelligence | Refers to the simplest bidding method when players provide bids and asks randomly. This is done mostly without any previous background regarding the performance of the market | [25,26,28,38,39] |
| Zero intelligence plus | The case when the bidding is performed according to the previous performance of the market, something like human behavior in the stock market. | [23,38] |
| Game theoretic bidding strategy | Sellers and buyers are modeled in a game that has two or more players. Each game player tries to win the game by making the optimal decision. | [3,37,40–44] |
| Adaptive Aggressiveness | Throughout the learning approach, quotations are automatically adjusted by market players based on the price of the market. | [39,45,46] |
| Inversed-Production Pricing | The prosumer predicts produced energy of their devices for 15 min intervals depending on data collected from their historical performance, then setting a price according to the relation of supply and demand. | [47] |
| Intelligently bidding agents | The agent's action is intelligent and makes bidding decisions based on reinforcement learning. | [36,48–57] |
| Parallel Multidimensional willingness | Multidimensional variables such as the historical records of trading data and counter behavior are modeled to mimic the microgrid fluctuation during bidding processes. | [46] |
| Prediction–integration | The past recorded transaction data are used by extreme machine learning that figures out the response of the electricity market to the prosumer's bidding. | [38] |

### 4.1.1. Comparable Bidding Methods

In the Continuous Double Auction (CDA), three bidding strategies are mostly used: zero intelligence (ZI), zero intelligence plus (ZIP), and adaptive aggressiveness (AA). The simplest bidding strategy is ZI that generates random prices, which is the commonly used method in the energy market of smart grid [38]. According to [58], ZI is the most appropriate bidding strategy in the CDA market. ZIP agent is proposed in [23]. In this type of bidding,

players can change their profit margin according to the earlier orders. The article [45] proposed the AA strategy in the blockchain-based electricity trading market. AA strategy refers to the state when agents' quotations are adjusted automatically using a learning approach based on the variation of the market price, which makes AA perform better than ZIP in terms of adapting and trading efficiency. The average efficiency of the AA strategy reached over 98%. This strategy can increase participants' profit compared to ZI & ZIP [45]. In CDA bidding, there is insufficient flexibility in changing the bidding amount. It creates a multidimensional willingness bidding strategy depending on historically recorded trading data, the connection between supply and demand, and offers flexible and rational bidding options. The P2P energy trading mechanism that is proposed is proficient in increasing the microgrids' incomes [46]. In order to maximize market efficiency, the agents need to be intelligent in the trading process. An intelligent bidding agent is employed and compared with ZI in terms of the agent behavior as in [5], which shows that the overall electricity price decreases while this type of bidding is implemented.

Bids and asks can also be provided based on the game-theoretical approach. In this approach, two or more market players can bid based on the price and other participants' behavior in earlier hours to bid below or above this price [37]. The article [41] highlights a complete P2P energy trading process, considering the integration of the Bayesian game bidding strategy with the physical constraints that occurred during the energy trade. A so-called Honestly bidding has been shown in [59] to be the dominant strategy for agents in the market, allowing the proposed technique to be evolved into the" bid-and-forget" market. Based on battery state-of-charge (SoC), a bidding strategy is created using the historical data of the system behavior [60]. An evaluation and comparison between game theory, ZI, and inverse-production pricing approaches were conducted in [46]. The game theory was the most privileged strategy that enables renewable resources to be the dominant supplier in the local energy market.

Further, ZI is mainly employed as a basic bidding strategy which provides the simplest method that enables traders to bid and offer prices randomly regardless of the other's decisions in the market. The lowest system efficiency has been provided by applying ZI approach [25,26]. ZIP is superior to ZI in terms of system efficiency when bidding is done within a specified maximum and minimum price margin determined according to the previous orders. This limit helps to estimate the predicted profits more efficiently than ZI strategy [23,38]. Game theory bidding strategy provides the feature of determining the deviation of lower and higher prices of the previous orders. According to [37], the best-offer game-theoretical bidding strategy was nearly ideal in a specific PV penetration. In [47], the game-theoretical approach is compared with Inversed-Production and ZI approaches, and the results found that the game theoretical approach is the best choice when it comes to supplying households with locally generated renewable resources.

### 4.1.2. Learning-Based Bidding Methods

Developing intelligent bidding and well-behaved agents is still a challenge in P2P energy trading. In different studies, 'cognitively bidding agents' were implemented, in which agents can vary the prices several times as introduced in [36]. Unlike other bidding strategies, agents can actively participate and create an interactive local market in this strategy. According to [52], a learning-based peer is used to ensure successful negotiation, increase the quality of agreements, and reduce the possibility of negotiation failure. Moreover, developing an 'intelligent negotiation agent' is proposed in [51] to apply learning capability in making bidding decisions. 'Artificial intelligent agents' are suggested in [50] to establish a real-time settlement and smart contracts which ensure that money transactions are only done when the energy is securely transferred to the consumer.

To improve the intelligence of energy trading, state-of-the-art methods are used as bidding strategies such as Reinforcement Learning (RL) and are introduced in different articles which can replace or support the human decision. RL is an algorithm that is used to measure stochastic tasks. In P2P energy trading, the RL framework plays a significant role

in addressing the issue of making decisions to maximize agents' cumulative profits [56,57]. Chiu et al. [57] proposed a novel bidding algorithm based on multi-agent Q-learning (MAQL) which is a framework of RL. Their proposed method is a double-sided auction-based market coordinated by an aggregator and can be combined with any existing bidding approaches to minimize time costs. In this learning-based bidding, the aggregator's business model is unknown to the end-users so bidding strategies are developed accordingly. MAQL algorithm was found to be an efficient method superior to the previously discussed method in terms of addressing renewable generators' uncertainty production as well as maximizing the profits of both end-users and aggregators. Nevertheless, it requires a cost of time during the process of learning. Further, for these bidding strategies to be sensitive to the learning rate, they should not be below a reasonable threshold.

Another study [56] used the RL method, particularly deep Q learning, to develop a learning-based bidding generator to find the optimal bidding strategy. The article concluded that the performance of this method is improved by 20% while using this strategy compared to the historical ones. Moreover, looking for a suitable partner to trade with, in the local energy market, is a time-consuming process. Therefore, an indirect trading paradigm among customers by a retail energy broker is proposed in [55] in which RL strategy is used to build the market model.

Zang et al. [54] used the RL algorithm to control community energy storage (CES) in the local P2P energy market through enhancing the decision-making of prosumers. In this study, trading is done within two stages without and with CES. In the first stage, the real-time local market, prosumers trade without a community energy storage, and those who failed to achieve successful trading in the first stage will be supported by the energy trade supporter in the second stage, which is with CES. This approach finds that the profits that are achieved were near the maximum of the daily transaction forecast compared to the other trading strategies. Further, the Q-learning algorithm is used in [53] a Continuous Double Auction as a method of decision-making in trading energy amongst microgrids. Q-cube formwork is designed in order to express a Q-value distribution. As a result, the overall profits of microgrids increased compared to the traditional methods of P2P energy trading.

### 4.2. Market Clearing Approaches

Several studies pay significant attention to finding the optimal market clearing methodologies, which differ according to the network scale, the market structure, and the participants' behavior; for instance, distributed methods are used in large-scale markets. In contrast, auction-based methods are used in the local market, and game theoretic-based methods deal with players with conflicting desires and objectives [12]. Khorasany et al. [12] present the current works on prospective local energy market designs, introducing a classification of the objective of market-clearing and its methods. They categorized the market-clearing approaches into two types: distributed and auction-based methods. Diverse types of market-clearing approaches are shown in Table 2.

Further, according to Khorasany et al. [12], the distributed approach for market-clearing guarantee scalability and reduce the accounting and communication expenses. The article [51] uses distributed optimization methods for market-clearing as well. Yap et al. [61] propose a two-stage market-clearing model, solved using the "Linear Programming optimization approach". Alam and colleagues [29] provide a transparent and private clearing approach through "multi-bilateral economic dispatch" (MBED) that is solved by consensus-based optimization, distributed Relaxed Consensus + Innovation RCI. This solving approach makes algorithms more complex when compared to the centralized solving approach. In [51], Alam et al. used distributed optimization methods for market-clearing and proposed Pareto optimality to address the issue of unfairly distributed costs in the P2P framework. Unlike other studies that consider the optimization of individual profit, they examine the energy price effects on the overall cost in a microgrid; the proposed model can be used as a predictor of the agent's behavior and perform the cost analysis.

The performance of several P2P trading mechanisms, Double Auction (DA), Mid-Market Rate (MMR), and Supply and Demand Ratio (SDR), are assessed, and the model of prosumer's decision-making is suggested in [62]. The authors concluded that the SDR mechanism favors electricity buyers; in contrast, the DA mechanism benefits the sellers and agents with batteries. Therefore, the MMR mechanism is the best choice when assessing the power's flatness. In [63], SDR is used to estimate the dynamic internal price within a microgrid. A method was proposed to evaluate the performance of three different trading mechanisms of P2P: MMR, SDR, and bill sharing (BS) in [60], which concludes that the SDR mechanism outweighed the other two mechanisms in terms of performance under moderate level of PV penetration, and the performance of BS is like the traditional model. Hadiya et al. [64] discussed three pricing mechanisms: MMR, BS, and SDR, versus coalition game-theory-based model. All of which were compared on different indicators of the performance of an institution in India. In [65], three pricing mechanisms, MMR, BS, and auction-based strategy, are proposed, and the results show that the cost is reduced in auction-based and MMR paradigms more than BS.

According to [42], the game theory-based approach was found to be fairer and more effective than the other approaches if peers in the energy market have conflicting interests. Yap et al. [61] compared the prosumers' revenues when clearing the market through motivational game approach or employing NEM, and game theory was more applicable and more profitable than NEM. Another type of market-clearing approach, so-called average mechanism is used in [66] which takes the average of bids and asks for this as the market price. This method is used with Double Auction in [18], and it is considered a scalable method, and all participants can take place in determining the price of the clearing market.

**Table 2.** Types of market-clearing methods as introduced in the literature.

| Market Clearing Method | References |
|---|---|
| Double Auction (DA) | [36,39,59,62,67–69] |
| Supply and Demand Ratio (SDR) | [60,62,63,67] |
| Mid-Market Rate (MMR) | [43,60,62,64,67,70] |
| Bill sharing (BS) | [60,65,67,71] |
| Game theory | [7,61,72–74] |
| Average mechanism | [18,37,75] |
| Pay-as-bid k-DA | [18,37,39] |
| Generalized second-price | [18,76] |
| Vickery–Clarke–Groves (VCG) | [18,37,41,77,78] |
| Trade reduction | [18,37] |
| Uniform price rule | [26,79,80] |
| Knapsack approximation | [81,82] |
| Greedy algorithm | [18,83,84] |
| Distributed optimization | [19,29,51,85–88] |

## 5. Game-Theoretic Approach

Game theory is defined as a mathematical tool used to analyze the behavior of different participants in a competitive environment and give the proper result. This model is used to provide a solution based on understanding the behavior of the other agents [70,89]. Countless articles introduced game theory as a useful tool to handle smart grid issues. According to [83], game theory is essential for decision-making research in the second generation of energy networks. Further, a detailed classification has been presented for recently used game theories and certain theoretic-auction models in the P2P energy exchanges.

### 5.1. Non-Cooperative Game Theory

This type of game theory is used to model participants with conflicting interests and make decisions without coordination or communication. This tool enables them to make decisions effectively and adequately. This category is further classified into static and

dynamic non-cooperative games [83]. The static type refers to participants' action when it happens only once without simultaneously knowing the other players' decisions. At the same time, the dynamic model indicates a case when the participants can repeatedly act according to the last actions [89]. Nash equilibrium is a popular non-cooperative game optimum solution that leads the non-cooperative game to a stable situation when players are not incentivized to deviate from the initial decision [1]. A non-cooperative energy trading mechanism for a centralized energy market among microgrids is designed in [46]. Bhatti et al. [40] propose a distribution level framework for energy trading by building up a non-cooperative, infinite strategy, and multiplayer game.

*5.2. Cooperative Game Theory*

This concept refers to a game when players cooperate to gain more profits from taking part. Its function is to estimate the number of players who intend to form a coalition in the game called Nash bargaining [66]. This type can be classified into three coalition forms: coalition graph game, coalition formation game, canonical coalition game [70,83]

1.  Coalition graph game handles communication between the participants and functions to derive low-complexity distributed algorithms for those who want to form a network graph. It also investigates the formed network properties, for example, stability and efficiency [83].
2.  The coalition formation game studies the network's structure, including adaptability, properties, and coalition cost.
3.  Canonical coalition game is the tool that distributes cooperation gains between players with fairness.

Cooperative game theory is used to change the selling price in cases where total electricity surplus is more than the total electricity deficiency, and the buying price if the total electricity surplus is less than the total electricity deficiency [90]. Based on the coalition game theory, the authors of [43] improve the mid-market rate (MMR) price model and introduce the weight variables. In [64], three pricing mechanisms MMR, BS, and SDR, are discussed and compared with the coalition game theory-based model.

Many studies of P2P projects pay significant concern to the importance of game theory in the future grid to build the business layer [7]. Several game theory strategies are presented in [7] to analyze the P2P energy market. The best-offer game-theoretic approach was found to be near the ideal efficiency [37]. Authors in [42] offer to model the trading mechanism and prosumer's decision procedure using game theory and Shapley value [42]. Tushar et al. [70] aimed in their article to increase the number of participants in the P2P network by using the motivational psychology system and developing a game-theoretic approach for the P2P energy market considering its specifications and categories, and for motivational purposes, game theory is found to be the proper choice that gives different models with certain properties. Hence, game theory is a promising modeling method if the designer considers prosumers' participation.

## 6. Auction Approach

An auction process is a market procedure based on negotiation techniques of the available bids to specify the buyer of the item according to the bidding's rules. Further, an auction can also be defined as "a well-specified negotiation mechanism mediated by an intermediary that can be considered as an automated set of rules" [12,91]. An auction approach for market-clearing in P2P energy trading system has been proposed in [92]. This study reviewed different types of auctions, and one auction approach has been designed for the applied platform. The implemented auction approach comprises three main levels: determination, allocation, and payment. It allows both the prosumers and consumers to trade energy via the platform by referring to specific rules without needing a third side to complete the trading process.

The authors of [8] have developed a P2P energy structure for a distribution system. A multi-round Double Auction has been utilized in constructing the proposed P2P system.

The proposed framework has been analyzed and tested on distribution networks of 33-node and 141-node.

Auction mechanisms have been widely implemented in different sectors in the power energy world. Microgrid, Smart-grid, Electric Vehicles, and other sectors are examples of the recent technology in the energy field that have utilized the auction approach frequently. In recent years, microgrid systems have been a topic of interest in the electrical energy world [93,94]. Many studies have been carried out to analyze and study the role of the MG as an energy system to supply energy to other homes or Microgrids. A P2P energy trading system for a MG network has been proposed in [65] to study three different market modes: mid-market rate, bill sharing, and auction-based pricing method. Further, an energy trading system has been proposed for Microgrids' energy auction based on blockchain mechanism [95]. This study also compares the proposed blockchain trading system with Vickrey–Clarke–Groves (VCG) auction mechanism.

Significant recent interest has also been directed to Electric Vehicle (EV) as a promising field in the energy systems' world. An energy-trading system among EVs is presented in [96]. An auction process is implemented to organize the trading of surplus energy between sellers and buyers. Authors have designed a naïve auction mechanism in which the auctioneer controls the energy trading auction to specify the prosumers' final price of the offered energy. A further study of the implementation of vehicles-to-grid technology on energy trading between EVs and grids is presented in [97]. The proposed energy trading system is modeled for two layers of vehicles to grid infrastructure: grid aggregator and EV aggregator layers. Each layer involved its auction mechanism. The numerical results indicate that the proposed auction systems enhance performance and minimize social costs.

### 6.1. Classification of Auctions

Much research has been conducted to analyze the classification of the applied auctions methods. Auctions can be classified into two main categories in the bids' rules: dynamic and single-round auctions [98]. Dynamic auction is an open auction in which multiple bids are submitted to the auction. During the dynamic auction, bidders can monitor the updates on the auction prices and modify their bids. Single-round auctions are defined as static auctions in which a single bid is submitted for each process, and this bid is a sealed one that is unknown by the other participants in the auction.

Dynamic auction has several pros and cons that affect its implementation. One of the main pros is reducing the 'winner curse' occurrence since the participants got the information about the pricing changes during the auction. The 'winner curse' is specified as the case where the winner of the item out of the auction has overestimated the item's true cost [99]. This case is indicated by the difference between the winning bid and the second bid that followed the winning one, which is represented by a significant difference between both of them. Authors of [100] studied and demonstrated the winner's curse effects on the industry field using a game theory technique. They specified the winner's curse level in two bidding frameworks: single-stage and multi-stage bids.

The main disadvantages of dynamic auctions are the complexity and long timings of the auction process, which may prevent low bids from entering the auction market. Meanwhile, the static auction is of less complexity, and it helps reduce the collision cases that may occur between the bids. However, the static auction has a common drawback represented by the possibility of low income for the item's owner because of the low bids submitted by the participants to reduce the possibility of the winner's curse occurrence.

An auction mechanism has been proposed for P2P energy systems to provide an energy trading system between the users connected to the network [44]. The proposed auction process of the energy trading system is based on a sealed-bid auction type. Different constraints have been followed through the auction process to meet an optimized method of the energy trading system. Examples of these constraints are energy operation cost, path infrastructure, transmission cost between peers, and available bids offers. Simulation results emphasize the ability of the proposed model to increase the profits of the participants.

Single-unit auctions can be composed of four basic types of auctions, which are [98]:

a.  Ascending-Bid Auction This type of auction is called an English auction, in which it starts its operation by specifying the starting price at the seller side. The starting price is considered a low price where it will be increased continuously until matching the available bids, in which the highest bid is the one who is granted the offered item.

b.  Descending-Bid Auction In a Descending-bid or Dutch auction, the starting price is considered a high price, and it decreases continuously until a bid matches the specified price.

c.  First-Price Sealed-Bid Auction In this type, the participating bidders present their bids once in a single-round auction in which they cannot modify their bids subsequently. After submitting all bids, the seller or auctioneer specifies the winning bidder who offered the highest bid, and the seller will grant a price that equals the highest bid.

d.  Second-Price Sealed-Bid Auction This type of auction is similar to the First-price sealed-bid auction except that the winning bidder will pay an amount equal to the second-highest available bid.

Auctions can also be classified based on the number of sellers and buyers per auction [101]. Auctions that involve one seller and multiple buyers or one buyer and multiple sellers are called Single-sided auctions. At the same time, auctions that are composed of multiple sellers and multiple buyers are called Double Auctions (DA). From the energy trading perspective, DA help in providing a two-sided market in which both sellers and buyers can switch their roles from offering their surplus power for the energy trading auction to buying it and vice versa. Single-sided auctions represented by one seller and multiple buyers are also known as Forwarding Auctions. On the other hand, single-sided Auctions with one buyer and multiple sellers are also known as Reverse Auctions [94].

DAs are represented as two-sided auctions since both sellers and buyers can submit asks and bids in case of their willingness to sell or buy items [102]. The sealed-bid Double Auction is one of the types of Double Auctions. It involves a single round of the auction process in which only the market operator knows the values of the submitted bids and asks [44].

The most common type of Double Auction is the Continuous Double Auction (CDA). In the CDA type, the bids and asks are submitted and sorted in descending (ascending) order to be matched and meet the clearing point [38]. Authors of [103] investigated and analyzed both the centralized and distributed energy trading markets in the CDA type. They also provided a comparison study between energy markets, representing the benefits for both consumers and consumers' sides, mainly characterized by electricity bills' reduction and decreasing the paying prices to be lower than the equilibrium prices. The combination between blockchain and the Continuous Double Auction is presented in [45].Periodical Double Auction (PDA) is another type of Double Auction. In this type of auction, the auction market is running and cleared in a predefined time called the market-clearing period. The buyers and sellers have to submit their bids and asks in the specified period. After that, the bids and asks are matched, transactions are created, and the market is cleared [104]. PDA implementation in the P2P energy trading system has been proposed in [104].

A Distributed Double Auction (DDA) has been proposed for P2P energy trading system in [105]. This type of auction indicates the ability of any connected peer to act as an auctioneer. Results show that DDA has more benefits in the case of energy transfer system more than the Centralized Auction type.

*6.2. Auction Mechanisms*

Auction processes are varied based on the type of applied auction mechanisms through the trading market. In the Sealed-bid Double Auction, after the buyers and sellers submit their prices, the total demand is specified by re-organizing the available bids from the highest available to the lowest. At the same time, the total supply is specified by sorting the available asks from the lowest one to the highest. The crossing point between the total

demand and the total supply represents the equilibrium price, which indicates the sold and purchased prices. The demand and supply curves are shown in Figure 3, where $p^e$ stands for the equilibrium price, bn is the available bids, and om is the available asks.

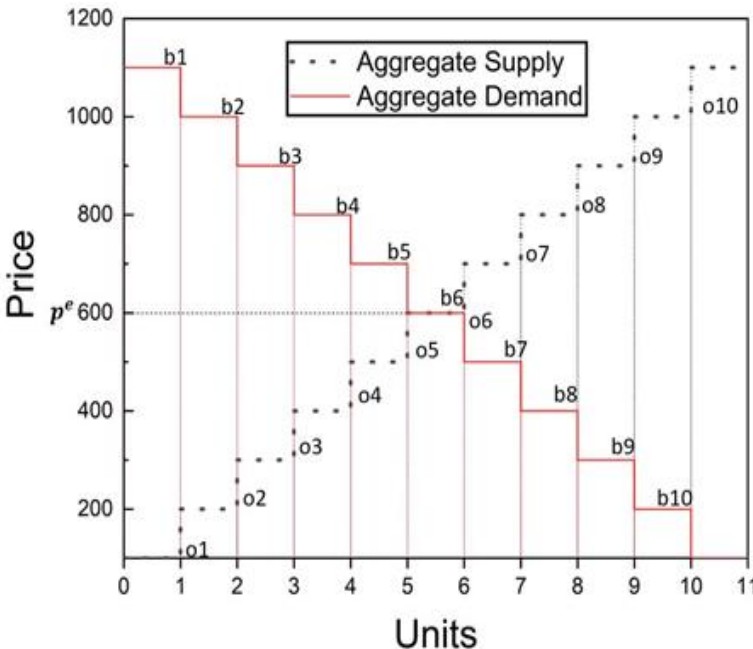

**Figure 3.** The operational principle of the Sealed-bid Double Auction.

Figure 3 represents the operational principle of the Sealed-bid Double Auction, which mainly involves three cases of the applied auction process. These cases are Sealed-Bid Double Auction with Uniform-Price Rule, Sealed-Bid Double Auction with Discriminatory Pricing Rule, and k-Double Auction.

In the case of the Uniform-Price Rule, buyers will pay the same price specified with the value of the equilibrium price, as represented by the shaded area in Figure 4a. The Discriminatory Pricing Rule depends on whether the exchange price is based on the bids from the buyer or the offers from the seller. With the rule of pay-buyer's price, the total payment is equal to the sum of all the winning bids as represented by the shaded area of Figure 4b. While in the case of pay-seller's price, the overall payment is specified by the summation of all the winning offers (as shown in Figure 4c).

The k-Double Auction is implemented in case of the intersection between the supply and demand for a range of price, not for a specific price as illustrated in the previous cases [106]. Applying this rule assures that bidders will not pay a price that exceeds their bids, and sellers will not obtain a price less than their asks. The following equation specifies the price out of this rule:

$$p = kb^*_{min} + (1 - k)o^*_{max} \tag{1}$$

Here, $k$ factor is defined as: $(0 < k < 1)$, is the lowest available bid above the offer and is the highest available offer below a bid.

Lin et al. [37] offered insights regarding the transactive energy market in P2P by studying several bidding strategies such as Pay-as-Bid k-Double Auction (k-DA) and Uniform k-DA and investigating their impacts on the conditions of the market. In the Uniform k-DA all players who win will participate at the same price. Pay-as-Bid k-Double Auction, also called discriminatory k-DA can outshine uniform k-DA in the average percentage of kWh sold, bought, and the average percentage of households cleared.

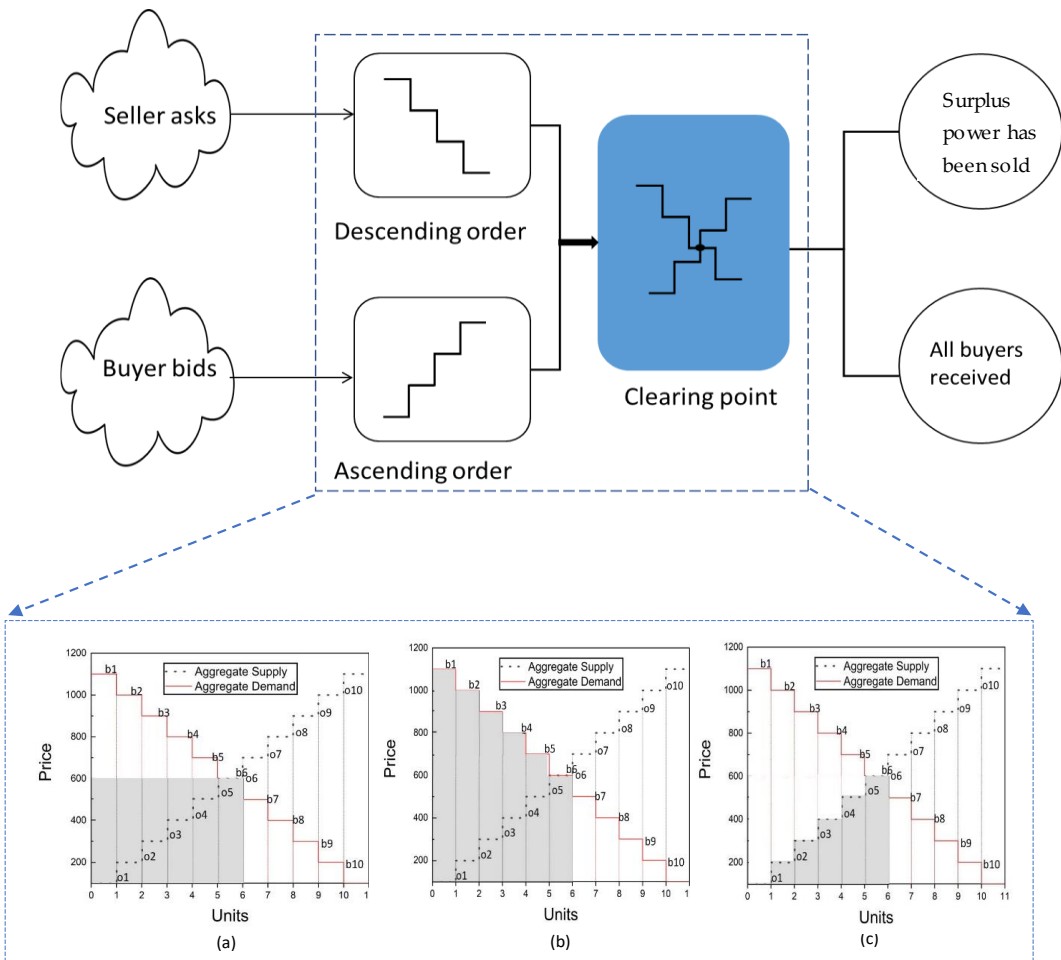

**Figure 4.** Auctions as a market-clearing approach (**a**) uniform price; (**b**) discriminatory pricing rule: pay-buyers' price; (**c**) discriminatory pricing rule: pay-sellers' price.

A greedy algorithm is applied as a market-clearing approach in [66], a scalable mechanism that allows buyers and sellers to participate in electricity pricing determination. In this method, the energy is transferred from sellers with the lowest prices to allocate to the buyers with higher bids. Knapsack approximation can be solved using a modified greedy algorithm, as introduced in [81]. It is used as a market-clearing mechanism for a single seller to various potential buyers. Every buyer suggests how much power they want and the price they are willing to pay, so energy sellers decide to allocate power.

In [59], the Double-Sided Auction is proposed as a market-clearing strategy for renewable resources, with a marginal price equal to zero. The average pricing market (APM) is proposed for zero marginal cost-pricing in the distributed renewable energy generation. To facilitate the operation and motivate prosumers to take part in the Continuous Double Auction (CDA) market by designing a data-driven market model with a prediction feature depending on Extreme Learning Machine (ELM)in which the CDA market is presented. Discriminatory k-DA can outshine and perform better than uniform k-DA in the average percentage of kWh sold and bought, as well as the average percentage of households cleared [26].

## 7. Business Models in Developing Countries

Most developing countries have insufficient electricity supplied to residents and limited access to electricity networks, especially in remote or rural areas where many live below the poverty line. Residents in these regions need incentivizing policies to motivate those who can afford to set up their renewable energy resources and storage systems to

share energy with neighbors, and encourage small companies to invest in renewable energy projects so that all community members access electricity at reasonable costs and maximum profits. Various P2P energy trading projects are carried out in different developing countries, such as Malaysia, India, Nepal, Thailand, Kenya, and Bangladesh. Each project studies the possibility of setting up P2P projects differently: challenges, business models, benefits.

In Malaysia, most prosumers with solar energy have no storage systems, and after P2P trading, a deficit or extra energy is there. Yap et al. [61] introduce a design of a market-clearing mechanism, using motivational game theory, Linear Programming optimization approach, exclusively suits utility and energy prosumers in Malaysia. Whether net metering or P2P is implemented in Malaysia, the national utility company, which is called TNB, will earn less because the consumed energy in the grid comes first from the solar panels of prosumers and has priority, so TNB will not be affected once this type of trading is used, while prosumers would have extra income.

The article [107] provides a detailed study of the P2P energy market, taking Nepalese rural areas as a case study. It discusses the constraints and barriers of microgrids and mini-grids and the potential enhancements. The technologies in Nepal are eligible to be applied to run micro and mini networks if certain obstacles such as system stability, political intervention, lack of awareness about renewable resources, and financial conflicts are addressed. Moreover, a transparent and secure market needs to be ensured.

Three pricing models, MMR, BS, SDR, are compared with the coalition Game theory-based model, all of which are compared to different indicators of the performance of an institution in India [40]. In [108], bilateral negotiation, one-to-one, is introduced in this paper to set up a P2P energy trading framework, aiming at enabling automated negotiations through new official modeling of agents, buyers, and sellers, so that the electricity bill is reduced and electricity is covered for all regions of developing countries. They tested the model on developing countries such as India, sub-Saharan Africa, and southern Asia. In these countries, semi-urban or rural areas mostly have unreliable networks. Using automatic negotiation to build a P2P trading market and modeling the agents' negotiation enables them to determine their preferences and decide the electricity price and quantities. The experimental results of the case study conducted on India's network illustrate that modeling agents increase the opportunity of rural areas to access electricity at low costs and with high benefits. Several game theory strategies are presented in [7] to analyze the P2P energy market. It also discusses the possibilities and challenges of implementing the P2P market in the Indian scenario.

The authors in [46] proposed business model instructions and guides based on blockchain to suit Thailand's electricity grid. They presented the business model guidelines by the theory in disruptive innovation, the case in which a small company with limited capabilities can defy and exceed an existing one. According to them, using blockchain in the future would lower the cost to consumers, prosumers, and small-sized enterprises (SEMs) below that of large and medium-sized groups. Takkabutra et al. [109] provide an overview of the current challenges and methodologies of applying P2P energy trading, taking Thailand as a case study, recommending that in order to install a P2P energy market in Thailand, there are various notes to consider: (1) The importance of developing the existing traditional network in terms of installed devices and energy quality. (2) The fairness of being charged for buying and selling electricity. (3) Harmonizing the rules and regulations to suit the new market mechanism. The article [31] reviews the electricity market structure and business model of P2P energy trading in Thailand, concluding that the proper wheeling charge rate has not been defined yet. Developing countries tend to use low-cost energy sources, making a decentralized system an appropriate infrastructure.

Another study gives a socio-economic overview regarding the DC network based on prosumers with no integration with the grid, evaluating the P2P market throughout this network, taking Bangladesh as a case study to investigate the motives and challenges of prosumerism. As an example of this, setting up small microgrids so that prosumers can share their surplus electricity with homes in need in Bangladesh rural areas. Accordingly,

the surplus electricity of prosumers could be sold between peers at a higher price than the case of selling it to the electricity retailers [110].

The "Auction House" model is proposed in [111] as a trading mechanism and simulation on 50 rural homes in Kenya. In this trading solution, electricity sellers and buyers can trade within a certain timeslot for unlimited durations. Thomason et al. [112] studied the implementation of P2P to enable poor people in rural areas of developing countries to pay electricity costs by providing affordable prices.

Business models' frameworks for mini-grid in developing countries, using Nigeria as a case study, are proposed in [113], considering four business models: government, private, public-private, and community-based. Two frameworks were proposed, "Technique for Order Performance by Similarity to Ideal Solution" (TOPSIS) and "Weighted Aggregated Sum Product Assessment" (WASPAS). Interval Type-2 fuzzy Sets (IT2F) are used to analyze linguistic information with multiple criteria for the business model. With TOPSIS, the private business model is the most suitable, while community-based is the best and the most suitable business model with the WASPAS framework.

## 8. Future Work

To further analyze the included papers and provide a visual overview, a literature classification is presented in Figure 5. This figure is built based on the included papers, and they have been considered as the core of literature classification.

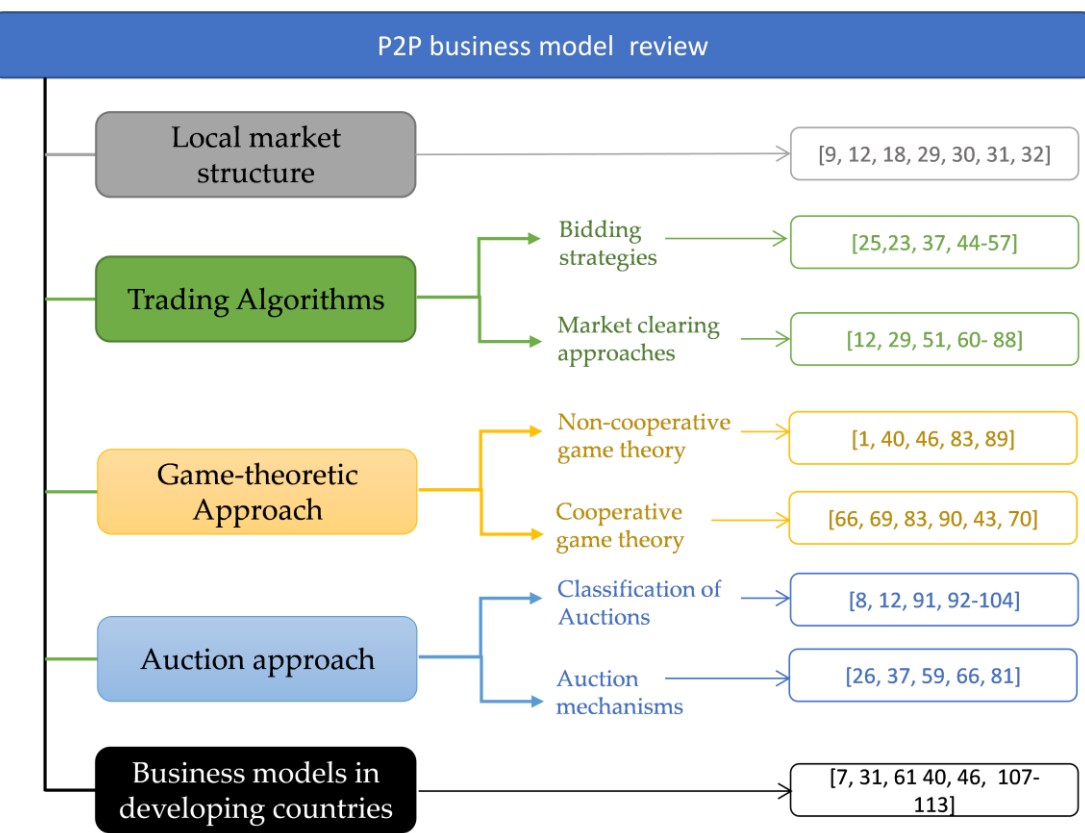

**Figure 5.** Literature classification [1,7–9,12,18,23,25,26,29–32,37,40,43–57,59–96,98–105,107–113].

Based on this figure and after reviewing in detail the listed papers, several research gaps, in terms of business models in P2P energy trading, are noticed, so intensive future research is needed to fill these gaps and enrich these certain fields with sufficient knowledge. Therefore, the proposed future research topics are as follows:

- Further work needs to be done to compare the P2P, community-based, and hybrid framework in the case of a larger number of participants to ensure fairness amongst players and employ optimal clearing algorithms, particularly when they communicate

asynchronously. Information exchange among players and building communication structures for players is another area of future research so that participants communicate with each other securely and feasibly.

- Socio-economic factors need further work to study the customer preferences and their energy demand, expressing human behavior, the extent of its rational decisions, and the effects of people's preferences on the performance of the market. This also should be conducted on the behavior of both prosumers and consumers so that the prosumer of high quality should be prioritized, and feeding households with low income prioritized as well. Further, the complex mutuality between social, political, and economic market objectives should be investigated to expand the market structure and bidding strategies.
- Distributed methods' scalability is still a heated issue that shall be addressed and enhanced to deal with an enormous number of players. So, research should be intensively focused on using these methods in building a trading platform for multiple buyers' and multiple sellers' negotiation. Moreover, diverse types of auctions and bidding strategies have to be evaluated to find the proper mechanism with extra attention to the intelligent bidding strategy, dynamic auction, and smart contracts to balance the P2P network under supply and demand variation.
- Further studies are essential to develop a suitable business model for developing countries regarding electricity pricing strategies for trading energy with restricted policies, particularly between nano-grids. This grid type is common in developing nations where strong mini-grids or microgrids are unavailable or weakly constructed.
- One of the main barriers to be addressed in developing countries is that centralized utilities control the whole market and refuse the installation of decentralized trading markets. So, to get existing utilities' cooperation, it must be ensured by extensive research that the proposed trading policies and trading strategies such as the P2P energy market will not adversely affect their profits. In addition, these traditional utilities also need to be taken into consideration, ensuring the viability of the energy trading market because players use their existing networks to transmit energy.

## 9. Conclusions

Distributed energy resources such as small-scale wind turbines and rooftop solar panels have become a vital component of the electricity grid. Integrating these resources with the grid needs proper policies to incentivize prosumers and companies to invest in renewable energy resources. P2P energy trading is introduced in the literature as an effective and profitable mechanism to accommodate the massive penetration of these resources. An exhaustive review is conducted on the business layer of the P2P trading mechanism in this article, explaining the trading algorithms, including a comprehensive summary of the different types of bidding strategies and market-clearing mechanisms. A detailed and systematic classification of auction mechanism and employing it in the energy market is uniquely presented in this paper. Developing countries face certain energy trading and management challenges due to constrained policies and limited capabilities. This paper surveyed the studies conducted concerning developing countries' networks and systems in terms of suggested P2P business models and trading policies. Finally, the business model structure is essential in developing an energy market with interactive features and adapting more renewable energy producers with motivated profits.

**Author Contributions:** Conceptualization, H.M., A.A.-H. and A.A. (Adib Allahham); methodology, H.M., A.A. (Adib Allahham) and M.A.-M.; Drafting, Resources, A.A. (Asma Alkhraibat) and M.H.; Writing, M.A.-M., A.A. (Asma Alkhraibat) and H.M.; Visualization, M.H., M.A.-M. and A.A. (Asma Alkhraibat); Project administration, H.M., A.A.-H. and A.A. (Adib Allahham). All authors have read and agreed to the published version of the manuscript.

**Funding:** This research was funded by [Royal Academy of Engineering], grant number [IAPP18/19-163, 2019].

**Acknowledgments:** The authors would like to thank the Royal Academy of Engineering for funding this project under grant Ref. IAPP18/19-163, 2019. Moreover, the authors would like to thank the Industrial Research & Development Fund in the Higher Council for Science & Technology for their support of the project.

**Conflicts of Interest:** The authors declare no conflict of interest.

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
