# Peer review of "Business Model of Peer-to-Peer Energy Trading: A Review of Literature"

_sustainability, doi:10.3390/su14031616_

Round 1
Reviewer 1 Report
The authors of “Business Model of Peer-to-Peer Energy Trading, A Review of Literature” made a comprehensive review of research on peer-to-peer energy trading and concluded their comprehension on the business layer of energy trading market. Substantial efforts were put on this manuscript.
The presentation of the submitted manuscript and the proposed approach requires some improvements before publication of the paper. I would like to see the effort of the authors to revise the paper addressing the following aspects:
- Obviously, “Peer-to-Peer” is the key word of this manuscript. This concept should be detailed reviewed on: What is a peer in the energy trading market? What is the general structure of peer-to-peer energy trading market? The concept of “business model” is mentioned in the title, and the position of business layer in energy trading model should be pointed out.
- Abbreviate should be detailed explained when the first time a long word appears in a paper, even if the word is a well-known one. (Line 83 DER)
- Figures in this manuscript are not appropriate. Figure 2 seems not academic enough in Line 209. Figure 1 is surely downloaded from other papers but not drawn by the authors in Line 488. Further, the numbers of figures are disordered.
- Reviews on the local market structure in section 3 should be more concise in summary. A chart or a graph showing the contrast of different approaches would be better.
- The classifications of bidding strategies and market clearing approaches in section 4.1 and 4.2 are the enumerating of different literature not a review, where should be paid more effort on.
- Section 9 is totally out of this manuscript’s concern. It should be deleted.
- As a review, a top-level design of the literature classification and key concern should be made before the list of papers. More efforts should be made.
Author Response
detailed response to the reviewer's comments are attached.

Reviewer 2 Report
The authors intend to cover various aspects of P2P energy trading; however, relevant studies were not properly compared and cited. The depth of this study must be improved significantly; otherwise, the audience cannot gain much insights into the state-of-the-arts.
For example, in Table 1, there is a category called ``Intelligently bidding agents,’’ but only 3 conference papers were cited, leaving out a number of journal papers without proper discussions. See, for example,
*State-of-the-art analysis and perspectives for peer-to-peer energy trading, Engineering, Elsevier
*Renewable energy bidding strategies using multiagent Q-learning in double-sided auctions, IEEE Systems Journal
What are the benefits of using RL as a bidding strategy? Also, what are possible drawbacks? When to use RL methods instead other approaches, such as game theoretic approaches? Those discussions are important; otherwise, simply a list of existing papers should not be considered as ``A Review of Literature’’
Author Response

(The authors gave the same response as above.)

Reviewer 3 Report
Energy security is one of the most important challenges and problems for the coming years, e.g. the rising prices of energy carriers in recent times. The presented literature review is an example of a scientific study, on the basis of which the authors give recommendations for further research and introduction of solutions to improve energy trading.
However, the article needs to be improved
Detailed comments:
Please check carefully the description on lines 76-81 - the described content and layout of the article is different from the actual one (there are 10 chapters); e.g. Research Methodolgy is in chapter 9 and not 7
The article uses different notation P2P, p2p and peet-to-peer, maybe it is worth to standardize
The article is chaotic, with partial definitions of objectives in different parts of the article.
The Research Methodology chapter should be after the Introduction section
Line 292 - should be Table 2
Line 489 - should be Figure 3
Author Response

(The authors gave the same response as above.)

Round 2
Reviewer 2 Report
The authors have addressed my comments well. A new segment regarding learning-based bidding methods has been added. Future work has been mentioned. Some terminologies and concepts have been discussed and clarified in Introduction. All changes were clearly marked. This reviewer thus suggests the publication of this work.